# *Candida albicans* Induces Oral Microbial Dysbiosis and Promotes Oral Diseases

**DOI:** 10.3390/microorganisms12112138

**Published:** 2024-10-24

**Authors:** Bina Kashyap, Sridhar Reddy Padala, Gaganjot Kaur, Arja Kullaa

**Affiliations:** 1Institute of Dentistry, University of Eastern Finland, 70211 Kuopio, Finland; arja.kullaa@uef.fi; 2Agartala Government Dental College, West Tripura, Agartala 799001, India; padalasri@yahoo.co.in; 3Shaheed Kartar Singh Sarabha Dental College & Hospital, Ludhiana 141105, India; doctorgaganjotkaur@gmail.com

**Keywords:** Candida, microorganism, oral microbiota, dysbiosis, oral diseases

## Abstract

*Candida albicans* are ubiquitous fungal organisms that colonize the oral cavity of healthy individuals without causing disease. *C. albicans* is an opportunistic microorganism with several virulent factors that influence the inflammatory process and allow it to invade tissues, evade host defense mechanisms, and release toxins, facilitating proliferation and degradation. At present, increasing emphasis is placed on polymicrobial interactions between *C. albicans* and various bacterial pathogens. Such interaction is mutually beneficial for both parties: it is competitive and antagonistic. Their complex interaction and colonization in the oral cavity serve as the basis for several oral diseases. The dispersion of *C. albicans* in saliva and the systemic circulation is noted in association with other bacterial populations, suggesting their virulence in causing disease. Hence, it is necessary to understand fungal–bacterial interactions for early detection and the development of novel therapeutic strategies to treat oral diseases. In this paper, we review the mutualistic interaction of *C. albicans* in oral biofilm formation and polymicrobial interactions in oral diseases. In addition, *C. albicans* virulence in causing biofilm-related oral diseases and its presence in saliva are discussed.

## 1. Introduction

The human oral cavity comprises host–bacteria complex oral microbiota, with interactive polymicrobial communities involved in oral biofilm formation [1]. The oral cavity exhibits broad diversity, with more than 700 microbial species having been detected, and it is the second primary site for microbial colonization after the colon. Most microorganisms in the oral cavity are bacteria; however, fungi, archaea, protozoa, and bacteriophages are also present. Oral health is balanced by symbiotic, competitive, and antagonistic microbiota activity [2]. Oral dysbiosis occurs due to disturbed interaction between inter-bacterial communities and host–bacteria, resulting in oral diseases such as dental caries, periodontitis, oral premalignancy, and oral cancer [3].

Cross-kingdom interactions between commensal fungi and oral bacteria require attention. The extensive interaction between fungi, mainly *Candida albicans*, and oral bacteria is crucial for their persistence and possibly contributes to infection [4]. It has been shown that in such interactions, fungi provide mechanical support for adhesion and colonization and provide an ideal substratum for bacterial attachment [5]. In diseases such as dental caries and periodontitis, *C. albicans* has been co-isolated alongside other oral microbial communities [6,7]. Interactions between bacterial species and *C. albicans* form biofilms through physical interactions, quorum sensing, and secretory proteases [8]. The microbial species in the oral biofilm release their cellular components and breach host barriers to reach distant sites and cause tissue destruction. Proteomic analyses of biofilms have shown virulence-related proteins among other pathologic secreted proteins [9].

The oral cavity is a complex organ with different niches colonized by millions of microorganisms. Several factors, such as a weak immune system, nutritional deficiencies, metabolic diseases, different types of diet (high sugar consumption), and poor oral hygiene and habits, can contribute to microbiome diversity, which facilitates the incorporation of Candida species into different oral niches [10]. The results of a recent study show that salivary change in smokers causes yeast overgrowth. The further spread of microorganisms via the bloodstream to different parts of the body occurs when infections are left untreated. The consequences of systemic infection have been reported in the literature, with mortality rates of 30–80% being noted [11]. The notable contributing Candida species include *C. albicans*, *C. tropicalis*, *C. glabrata*, *C. krusei*, *C. parapsilosis*, and the recently identified *C. auris*. Most Candida infections are endogenous, apart from *C. krusei*, which has an exogenous source [12]. It has been shown that *C. albicans* and its biofilms can induce cytokine production in host cells. The biofilm of *C. albicans* provides a suitable location for bacteria, and during such mutual co-existence, microbial cells avoid host recognition or elimination through antimicrobial action [13]. Hence, *C. albicans* can change the oral microbial environment, become pathologic, and induce cell–cell interaction with bacteria. This narrative review focuses on *C. albicans* mutualistic interactions with oral commensal bacteria and its change in different oral diseases. In addition, the cross-kingdom interaction between Candida and bacteria and its impact on polymicrobial biofilm and the host are presented. We also highlight the relevance of saliva and salivary metabolites to biofilm-related oral diseases caused by *C. albicans*.

## 2. *C. albicans* in Oral Biofilm Formation

Biofilms are organized societies of various microbes, embedded within self-made extracellular polysaccharides. Biological macromolecules such as proteins, carbohydrates, and nucleic acids comprise the matrix scaffold. Such biofilms are observed practically everywhere on humid surfaces, including the oral cavity [14]. The development of oral biofilm is a dynamic process where (i) a planktonic microbe attaches to a surface randomly or via chemical attraction, (ii) the microorganisms join to form microcolonies in the complex biofilm, and (iii) the virulent microbial colonies disperse and colonize distant favorable areas [15]. *C. albicans* biofilms contain yeast, pseudo-hyphal, and hyphal-form cells. After initial adhesion, the biofilm matures with increased cell proliferation and morphological change from yeast to hyphae [16]. With further development, a biofilm matrix forms that supports the cells in unfavorable environmental conditions. The biofilm matrix of *C. albicans* contains protein, carbohydrates (mainly mannans and glucans), lipids, and nucleic acids (representing approximately 55%, 25%, 15%, and 5% of the biofilm matrix). Fungal hyphae are fundamental components that support and provide a scaffold for the attachment of additional yeast cells as well as bacteria, developing multispecies biofilms [17]. In the final phase, a mature biofilm is formed, and the diffusion of nonadherent cells occurs, resulting in the dissemination of fungal cells in the tissue. Most of the dispersal occurs from the uppermost layers of the biofilm [18] (Figure 1).

The adhesion of oral microbes to host tissue is required for tissue invasion and infection. *C. albicans* adhesins play a major role in biofilm formation. The three important adhesion families are (a) the agglutinin-like sequence (Als) family, (b) the hyphal wall protein (Hwp) family, (c) and the individual protein file family F/hyphally regulated (Iff/Hyr) family [19]. Among these families, during biofilm formation, the Als family is important in mediating the initial attachment of *C. albicans* yeast cells. These yeast cells enable adhesion via an amyloid-forming region, critical for yeast cell–cell aggregation and cell-substrate adhesion [20]. Extracellular deoxyribonucleic acid (eDNA) released by *C. albicans* contributes to biofilms’ development, maintenance, and stability. eDNA can modulate the host immune response by activating bone marrow-derived myeloid dendritic cells [21]. Similarly, neuregulin 1 (Nrg1), pescadillo (Pes1), and unscheduled meiotic gene expression (Ume6) are known *C. albicans* transcription regulators of dispersal. The results of an in vitro study showed the overexpression of Nrg1, which promotes the dispersal of yeast cells from the biofilm. In addition, Nrg1 is a negative regulator of the morphological switch from yeast to hyphae [22]. The overexpression of Pes1 also indicates increased dispersal; in comparison, the overexpression of Ume6 reduces the dispersal of fungal cells. Moreover, dispersed cells showed enhanced pathogenicity in an in vivo study [18]. Other transcriptional factors contributing to *C. albicans* biofilm formation, development, and dispersion are presented in Table 1 [23,24,25,26,27,28,29,30,31,32,33,34,35,36,37,38].

## 3. *C. albicans* Virulence and Host Cell Response

Over the past few decades, it has been established that *C. albicans* can become pathogenic under certain conditions. Changes in the properties and morphological variations of *C. albicans* are indicative of its virulence. Complex regulatory networks, signaling pathways, and environmental changes are responsible for the transition of yeast-like cells to hyphae [39]. Active penetration into host cells by the invasive and pathogenic *C. albicans* hyphae is regulated by the cAMP-dependent protein kinase A pathway [40]. The scaffold formed by *C. albicans* allows several bacteria to form multispecies consortia, which are heterogeneous, multicellular, and multilayered three-dimensional structures bounded by an extracellular matrix (ECM) [41]. The members of the adhesins family of *C. albicans* interact with specific host ligands, such as laminin, fibronectin, collagen, fibrinogen, vitronectin, or complement proteins [42]. Other virulent factors include enzymes (proteases, phospholipases, and lipases), toxins (hypha-specific α-helical amphipathic peptide—candidalysin), and molecules involved in quorum sensing (farnesol, farnesoic acid, phenylethyl alcohol, tryptophol, and tyrosol). All of these substances facilitate penetration into the host tissue, triggering cellular stress and host inflammatory responses [43].

The human oral cavity is guarded by the oral mucosal barrier complex, the oral mucosal immune system, and salivary defense mechanisms. All play a key role in the oral balance between health and disease [44]. *C. albicans* expresse pathogen-associated molecular patterns (PAMPs) on their surface that are recognized by the pathogen-recognition receptors (PRRs) of oral epithelial cells [45]. This process orchestrates host cytokines through signaling molecules and activates the innate and adaptive immune systems. In addition, salivary secretory IgA, in the first instance, suppresses harmful fungal effectors and conversely, interacts with bacteria to maintain balance in the commensal microbe’s composition [46]. Oral mucosal epithelial cells exhibit toll-like receptors (TLRs) at the cell surface for the recognition of fungal mannoproteins and the detection of fungal DNA. This recognition leads to the release of pro-inflammatory cytokines and chemokines that promote macrophages and neutrophils to the infected area [47]. Oral epithelial cells are sensitive to the fungal toxin, candidalysin, as it can activate the epithelial growth factor receptor (EGFR), leading to several responses [48]. However, phagocytes play a crucial role in mucosal homeostasis against fungal dysbiosis [49].

## 4. *C. albicans* and Oral Bacteria—A Cross-Kingdom Interaction

The oral cavity comprises unique anatomical features of soft (mucosal) and mineralized hard (tooth) tissues for Candida colonization. *C. albicans* primarily affects the soft oral mucosa rather than teeth. However, the niches around the tooth surface are the preferred site for dental biofilm (plaque) formation intermixed with *C. albicans* and bacterial species [50]. Cell–cell adhesion is the main factor that mediates *C. albicans* and oral bacterial species interaction. The *C. albicans* cell wall contains polysaccharides, including glucans, mannans, chitin, and several adhesion proteins and receptors that help it interact with oral microorganisms, forming a polymicrobial biofilm. In this polymicrobial biofilm, *C. albicans* grows along with Gram-positive and Gram-negative bacteria [51]. The interaction between *C. albicans* and mitis group streptococci, (such as *Streptococcus gordonii*, *Streptococcus sanguinis*, and *Streptococcus oralis*), *Porphyromonas gingivalis*, and *Staphylococcus aureus* forms a well-organized structure [52]. *S. oralis* and *S. gordonii* have been observed forming near the *C. albicans* hyphae within a few hours of nutrient supply [53].

*S. gordonii*, an oral commensal non-pathogenic bacterium, plays an integral role in dental plaque formation. It was shown in an in vitro study of early polymicrobial biofilm formation that *S. gordonii* binds to *C. albicans* via cell wall proteins and enhances hyphal development [54]. In addition, the O-mannosylation of the *C. albicans* cell wall contributes to the development of an inter-kingdom biofilm [55]. *Streptococcus mutans*, the main cariogenic bacteria work in coordination with *C. albicans* and other mitis group streptococci in a sucrose-dependent partnership. Rather than cell–cell adhesion, *S. mutans* produce extracellular polysaccharides (water-insoluble glucans) to establish interaction with *C. albicans* [56]. In the biofilm containing *C. albicans/S. mutans*, *S. mutans* form microcolonies around fungal cells entangled in an extra polysaccharide-rich extracellular matrix [52]. The results of a recent study have confirmed that in the absence of a sucrose environment, C. albicans binds twice as strongly to *S. gordonii*, and in the presence of a sucrose environment, the binding of *S. mutans* and *C. albicans* increases dramatically (up to ∼6 fold) [57].

Furthermore, extracellular signaling, quorum sensing molecules, and other factors can facilitate *C. albicans* interaction with oral bacteria (Table 2) [20,54,58,59,60,61].

It is recognized that the quorum-sensing molecule, farnesol, in *C. albicans* promotes yeast-to-hyphae transition. Recently, in a *C. albicans/S. mutans* biofilm, low levels of farnesol were shown to stimulate glucosyltransferase-I (GtfB) expression/activity and increase bacterial growth. In contrast, high levels of farnesol were found to inhibit *S. mutans* growth [61]. Cross-feeding in biofilm maintains the specific environment for the growth of microorganisms. For example, *S. mutans* breaks sucrose into glucose and fructose, where glucose is used as a carbon source by *C. albicans* and enables persistence in acidic conditions. *S. mutans* and *C. albicans* biofilms produce lactate, formate, and fumarate as carbohydrate metabolism products, which facilitate their growth and make them acid tolerant [62]. *C. albicans* ensures strictly anaerobic conditions by consuming oxygen from the local environment and increases the abundance of anaerobic bacteria of the *Veillonella*, *Prevotella*, *Leptotrichia*, and *Fusobacterium* genera [63]. As an example, *C. albicans* can provide a hypoxic microenvironment to support the growth of *Bacteroides fragilis*, *B. vulgatus*, and *Clostridium perfringens* [64]. Hence, the commensal *C. albicans* growth in the oral cavity that enhances polymicrobial interactions, cross-feeding, and environmental change facilitates the growth of several pathogenic oral microbiomes and causes oral microbial dysbiosis.

## 5. *C. albicans* and Oral Bacteria Interactions in Oral Diseases

Candida spp. are found in mucous membranes, such as the gastrointestinal tract, mouth, nose, reproductive organs, skin, etc. *C. albicans* is a member of the resident human microbiota and is responsible for several types of oral diseases [65]. The dispersion of candidal cells is ‘preconditioned’ for maximum virulence where it upregulates the expression of genes whose products are involved in the acquisition of micronutrients, drug resistance, and the hydrolysis of host substrates [66] (Figure 2). Table 3 presents several studies confirming the overpopulation of *C. albicans* in the saliva of patients with oral diseases where *C. albicans* increases the risk of disease progression [67,68,69,70,71,72,73,74,75,76,77,78,79,80,81,82].

### 5.1. Dental Caries

In dental caries, *C. albicans* (yeast cell wall) adhesins interact with the salivary pellicle and adhere to the tooth enamel. *S. mutants* are the main causative organism of dental caries; they selectively bind to salivary lectins in the pellicle. After binding, *S. mutants* and other microbes proliferate to form a three-dimensional community to distribute nutrients, oxygen, and signaling molecules [83]. It has been shown that children with oral *C. albicans* are at increased risk of dental caries (>5 times) compared to children without *C. albicans* [84]. In the oral cavity, *C. albicans* can produce acid by metabolizing carbohydrates (i.e., glucose) to cause a reduction in pH from 7 to 4 [85]. In an in vivo study, it was shown that *C. albicans* has a roughly 20 times stronger ability to dissolve hydroxyapatite than *S. mutans*. *C. albicans* adheres to hydroxyapatite through electrostatic interaction [86]. *C. albicans* colonization occurs through the mycelial network with bacteria or by forming a spatial arrangement with *Streptococcus* [87]. The total loads of *C. albicans* and *S. mutans* were increased in supragingival dental plaque obtained from active carious lesions in children with early childhood caries [88]. *C. albicans* was isolated with *Peptostreptococcus micros* from root canal samples in patients with persistent endodontic infections. These findings implicate Candida in root canal infections with pulp necrosis [89]. *C. albicans* raises the glucose intake of *S. mutans*, which was confirmed when *S. mutans* were co-cultured with *C. albicans*. The co-culture showed a higher glucose metabolic rate than the pure-culture group. Furthermore, the transcription of *S. mutans* genes associated with the transportation and metabolization of carbohydrates is affected by *C. albicans* [62,71,90]. During coexistence, *S. mutans* genes are upregulated and participate in carbohydrate metabolism, galactose metabolism, and glycolysis/gluconeogenesis. Similarly, *C. albicans* genes responsible for carbohydrate metabolism (sugar transport, aerobic respiration, pyruvate breakdown, and the glyoxylate cycle) are enhanced by co-culturing [62].

Of late, findings on *S. mutans* cells secreting membrane vesicles have been presented. Membrane vesicles have been shown to contain the glucosyltransferase-I (Gtf) enzyme that, in the extracellular matrix of *C. albicans* biofilm, can contribute to sucrose metabolism [60]. The results of a recent study showed that the protein kinase A (PKA) pathway plays a central role in *C. albicans* activity. Mutanocylin (unacylated tetramic acid) secretion has been shown to have an impact on the transcriptional profile of *C. albicans*, which mainly regulates cell wall biogenesis and remodeling through the PKA signaling pathway [91]. The presence of *C. albicans* and *S. mutans* in dental caries produces synergistic and antagonistic effects. The synergistic effects during biofilm formation, through quorum sensing molecules, and in metabolic activity are discussed above. The results of salivary studies have also shown the prevalence of *C. albicans* in dental caries (Table 3).

### 5.2. Periodontal Disease

Periodontitis is a polymicrobial disease that affects the supporting structures of the teeth, causing attachment loss and bone loss. It is caused by a synergistic and dysbiotic dental plaque microbial community that results in disruption of tissue homeostasis. The main pathogens involved are *Porphyromonas gingivalis* [92]. *Tannerella forsythia*, and *Treponema denticola* also colonize with *P. gingivalis*. *P. gingivalis* is known to induce inflammation by remodeling the microbiota from a normal state to dysbiosis [93]. Periodontal microbial progression is mediated by a change in pH or redox potential, or a decrease in oxygen level. This change facilitates the existence of the subsequent colonizer. Lastly, the close intercellular interaction engages the microbial surface adhesins on the periodontium [94].

*C. albicans* and early colonizers such as mitis group streptococci form a scaffold for other microbes to attach. *Fusobacterium nucleatum* is a bridging colonizer between fungal biofilm and periodontal pathogens in the oral cavity. The results of an extensive in vitro study on *F. nucleatum* showed that it is able to inhibit *C. albicans* growth and filament formation without affecting its cell viability [95]. It is hypothesized that if the fungal morphological changes are blocked by bacteria, this weakens the host immune response. As a result, it is beneficial for them to remain unnoticed, escape the host immune system, and spread to other organs [95]. Other bridging colonizers include *P. nigrescens* (genus *Prevotella*) and *Campylobacter*. *P. nigrescens* can modulate fungal biofilm formation and fungal cell viability. The viability of *C. albicans* decreases with the increase in *P. nigrescens* cell abundance [96]. A similar effect was observed in a study involving *Campylobacter* where secretion of bacteriocin-like substances inhibited the growth of *C. albicans* [97].

*P. gingivalis*, *T. forsythia*, and *T. denticola* belong to the red complex as they are the major etiologic agents of periodontal disease. *P. gingivalis* induces germ-tube formation of *C. albicans* in both oral isolates and the strain. *P. gingivalis* results in the generation of a more invasive fungal phenotype [98]. *P. gingivalis* effortlessly adheres to the blastospore or pseudohyphae form of *C. albicans* and this adhesion is mediated by the bacterial internalin protein family (InlJ) and gingipain, Arg-x-specific proteinase, and adhesins (RgpA) [99]. In a study, chronic periodontitis patients showed higher levels of Candida colonization compared to healthy controls; however, the relationship between Candida colonization and the severity of chronic periodontitis could not be established [79]. Nonetheless, it is recognized that *C. albicans* play a role in the formation of periodontal microbial plaque and the adherence of bacterial species to the periodontal tissues.

Oral microbiome dysbiosis in periodontal diseases can be assessed using saliva and salivary metabolomics (Table 3). Upregulation in the salivary levels of valine, isoleucine, phenylalanine, tyrosine, proline, succinate, butyrate, and cadaverine was presented in our previous publication [44]. The synthesis of polyamine biosynthetic enzymes, such as ornithine decarboxylase and spermidine synthase, also depends on the presence of Candida yeasts in polymicrobial biofilm [100]. In addition, changes in the salivary levels of lactate, pyruvate, N-acetyl groups, and methanol are indicators of oral health or disease. Changes in lactate and pyruvate levels can influence the cell wall of *C. albicans* and its proteome and secretome, depending on access to carbon sources [101].

### 5.3. Oral Precancer

Interest in the study of *C. albicans* has increased because of its association with precancerous lesions of the oral mucosa. In 2007, the World Health Organization (WHO) proposed the term oral potentially malignant disorders (OPMD) for precancerous lesions and conditions [102]. Candidal leukoplakia is a precancerous lesion, with a high rate of malignant change [103]. For decades, however, it has been debatable as to whether Candida species are secondarily acting on a pre-existing leukoplakia or leukoplakia’s high malignancy rate supports the role of fungus. The results of prior animal studies have shown epithelial hyperplasia and cellular atypia induced by Candida species [104].

Candida is known to produce nitrosamines, namely, N-nitrosobenzylmethylamine, a carcinogenic compound. Candida strains obtained from advanced precancerous lesions showed dispersion of candidal infections from the superficial mucosal surface to the deeper epithelial cell layers, thereby transporting and depositing carcinogenic compounds into deeper layers and causing dysplasia [105]. It was postulated that these compounds may directly, or with other chemical carcinogens, activate specific proto-oncogenes to initiate the development of a malignant lesion [106]. A positive association between fungal infection and different grades of epithelial dysplasia, squamous cell carcinoma, and hyperkeratosis was observed; despite this, no causal relation between Candida species and other lesions was established. However, the presence of Candida and other microorganisms in the lesions was confirmed [107]. Hence, the relationship between *C. albicans* and oral precancer, and its malignant transformation, remains controversial. However, it is reasonable to postulate that *C. albicans* is a co-causative factor in oral precancers.

### 5.4. Oral Cancer

Recent statistics show that over 389,485 new cases of oral cancer and 188,230 deaths are reported annually [108]. Roughly, 90% of oral cancer cases are oral squamous cell carcinoma (OSCC). Various aggravating factors contribute to OSCC development, such as oral dysbiosis, genetic, and environmental factors, and exposure to chemicals, which can induce genetic and epigenetic alterations [109]. Current investigations are inclined toward the oral microbiota’s role in carcinogenesis primarily through chronic inflammation, the synthesis of carcinogens, and epithelial integrity change. The incidence rate of Candida infection reported in OSCC varies from 25% to 74.7% [110]. The formation of *C. albicans* hyphae results in the production of interleukin (IL1β) and activates proinflammatory cytokines. *C. albicans* colonization in OSCC shows genotypic diversity that affects the carcinogenic process [111].

Several molecular mechanisms are proposed with regard to dysplasia and malignancy induced by *C. albicans*. These mechanisms are as follows. (1) Candida upregulate proinflammatory cytokines, interleukins (IL), tumor necrosis factor (TNF)-α, interferon (IFN-γ), and the granulocyte monocyte colony-forming unit (GM-CSF). The cytokines influence the metabolic pathways and induce endothelial dysfunction. Hence, they promote cancer development by altering the host immune system [112,113]. (2) p53, a cell proliferative marker, was significantly more abundant in lesions exhibiting epithelial dysplasia with candidal infection [114]. p53 and Ki-67 overexpression is widely recognized in malignant lesions. Prostaglandin endoperoxide synthase 2 (COX-2) releases prostaglandins and increases inflammation in cancer. Thus, it influences cell proliferation, cell death, and tumor invasion [115]. (3) The release (via hyphal invasion) of nitrosamines, such as N-nitrosobenzyl methylamine, into the dysbiotic oral cavity, causes tumor development [105]. (4) The virulence factor of *C. albicans* is also by the production of acid aspartyl proteinase, which maintains an acidic pH and degrades the extracellular matrix (laminin 332 and E-cadherin). Therefore, dysplastic alterations in the oral epithelium and the dissemination of *C. albicans* into the systemic circulation occur [116]. (5) In oral cancers, *C. albicans* can produce large quantities of acetaldehyde and acetyl-CoA synthetase. Acetaldehydes are produced by metabolizing ethanol and glucose. Due to mutagenic qualities in DNA, acetaldehyde acts as a carcinogen [117,118]. (6) *C. albican* hyphae produce the toxin ‘candidalysin’ which work in conjugation with the cytolytic activity of *C. albicans*. Candidalysin can induce NF-κB and MAPK pathways and excite GM-CSF, an essential component in carcinogenesis. In a study, it was shown to induce mucosal epithelial damage and elicit host inflammatory processes by triggering NLR family pyrin domain-containing protein 3 (NLRP3) [119,120].

## 6. Discussion

*C. albicans* causes disease by altering the host’s immune condition and the pathogen. The change in the oral microbial population and preeminent growth and colonization of *C. albicans* on the oral mucosa cause it to develop into a disseminated form [121]. Factors that contribute to such an onset include prosthesis, salivary gland dysfunction, certain medications (i.e., broad-spectrum antibiotics and steroids), and a high-carbohydrate diet. Smoking cigarettes, diabetes, cancer, and immunosuppression are important contributing factors for *C. albicans* to aggravate oral diseases [122]. Mono-species colonization seldom causes infections of the oral cavity. As discussed, *C. albicans* is co-isolated alongside oral microbial communities in oral diseases. A high incidence of mixed colonization of *C. albicans* (66.7%) and *S. aureus* (49.5%) was shown in individuals with denture stomatitis. The cocolonization or coinfection of fungal–bacterial interaction (polymicrobial interaction) is associated with a multitude of other conditions including infections of endotracheal tubes, biliary stents, silicone voice, orthopedic prostheses, and acrylic dentures [123,124].

Concern arises when the mutual coexistence of pathogens within the biofilm and the local transformation of their environment occurs. In a study, an in vitro study model of a mixed biofilm formed by *P. gingivalis* and *C. albicans* showed inhibition of host cell migration. The combined effect of organisms appeared stronger than the separate effects [125]. Inter-genus interaction also occurs within mixed *C. albicans* bacteria biofilms. Although C. albicans is studied extensively, other candidal species such as *C. dubliniensis*, *C. glabrata*, *C. krusei*, *C. tropicalis*, and *C. auris* are also sources of increased concern in the clinical setting. In a study, dental acrylic resin strips showed mixed species biofilms in the oral cavity, predominantly, *C. albicans* and fewer non-*C. albicans* species [126]. In addition, in other studies, *C. albicans* and *C. dubliniensis* organisms were found in the oral cavity of immunocompromised patients, and patients with denture-associated stomatitis presented with *C. albicans* and *C. glabrata* [127,128]. Such mixed microbial species biofilms pose a major threat in the clinical setting due to issues relating to drug resistance.

Another important concern is the metabolic interactions in polymicrobial environments for microbial adaptation. Direct or indirect metabolic interactions between microbes are involved in the production of metabolites. The metabolites produced by one species are subsequently consumed by another or enable other species to persist [129]. Saliva is a complex biological fluid where salivary components serve as a defense against *C. albicans*. Levels of salivary metabolites such as histidine, tyrosine, choline, phosphoenolpyruvate, octanoate, uridine monophosphate, 6-phosphogluconate, ornithine, butyrate, aminovalerate, and aminolevulinate have been shown to increase or decrease in patients with Candida infection. The release of such metabolites affects *C. albican* growth, oral biofilm formation, pyrimidine synthesis, and nucleic acid synthesis [130]. The comprehensive analysis of salivary metabolites sheds light on their role in *C. albicans* pleomorphism. Hence, the metabolites observed in the saliva confirm oral dysbiosis and its association with oral inflammatory diseases and oral cancer.

It is a challenge to eradicate Candida–bacterial polymicrobial biofilm-induced oral diseases. Polysaccharides secreted by microorganisms in biofilm play an important role in preventing drug penetration and, hence, lead to antibiotic resistance [131]. The biofilm formed by *C. albicans*, and *S. aureus* cells can induce bursts of reactive oxygen species that activate the expression of the outflow pump in *S. aureus* cells and enhance drug tolerance [43]. Difficulties arise because antifungal therapy targeting only *C. albicans* is nonspecific. Therefore, a new approach to identifying effective therapeutic techniques for Candida–bacterial polymicrobial biofilm emphasizes inhibiting potential pathogens and the environmental factors that promote the selection and enrichment of the microbiota.

## 7. Conclusions

In conclusion, a strong association can be observed between *C. albicans* in oral biofilms, Candida–bacteria interactions, and oral diseases. It appears that *C. albicans* plays an important role in causing oral microbial dysbiosis, which could serve as the basis for various oral diseases. The salivary metabolic signature provides information on biomarkers specific to *C. albicans*, contributing to early detection and the establishment of new treatment methods. Most *C. albicans* biofilm research focuses on biofilm development, antifungal resistance, and polymicrobial interactions. However, how to effectively utilize this knowledge to develop efficient strategies to treat *C. albicans* biofilms and biofilm-related infections remains uncertain.

## Figures and Tables

**Figure 1 microorganisms-12-02138-f001:**
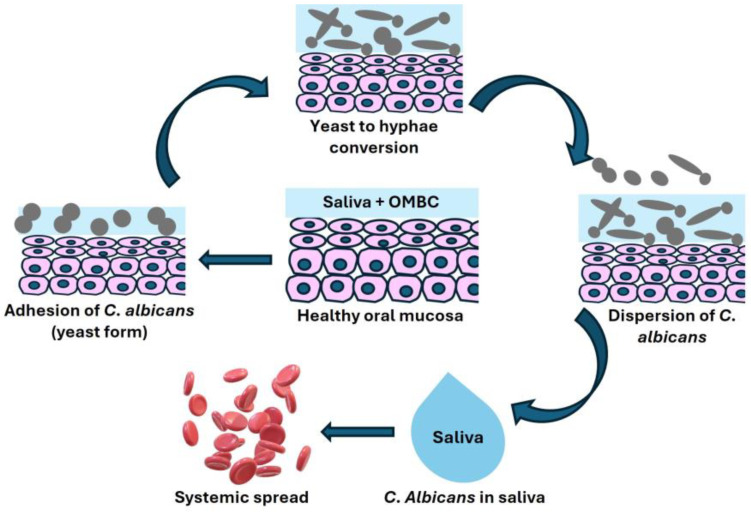
Change in the healthy oral mucosa to *C. albicans* oral biofilm formation following the stages of the initiation, adhesion, conversion, and dispersion of fungal organisms into the saliva and the systemic circulation.

**Figure 2 microorganisms-12-02138-f002:**
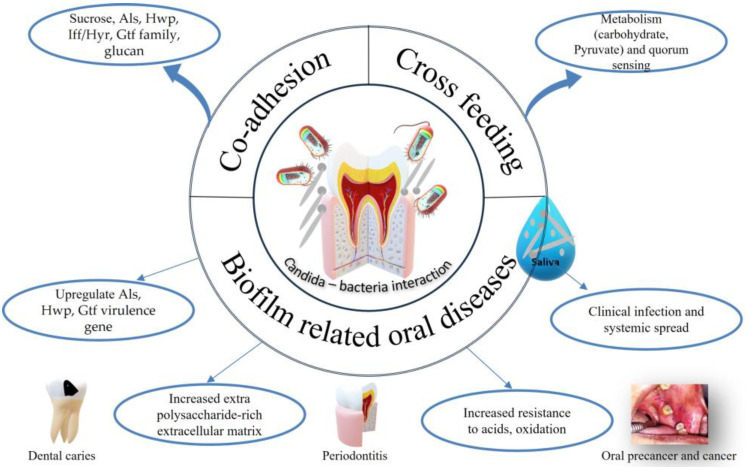
Schematic representation of Candida–bacteria interactions where Candida helps in co-adhesion via adhesion family. After co-adhesion, cross interaction metabolism occurs to supply nutrients for the growth of microbes and significantly increases biochemical activities. Later, virulent factors are upregulated, and more microbes accumulate to manifest several biofim-related oral diseases.

**Table 1 microorganisms-12-02138-t001:** Transcriptional factors contributing as positive and negative regulators to *C. albicans* biofilm organization formation, development, and dispersion.

Transcriptional Genes	Study Type	Function	References
BRG1—Brahma-related gene 1; BPR1—Backpressure regulator 1; FLO8—Flocculation 8; GZF3—Gata zinc finger protein 3; RFG1—Repressor of filamentous growth 1; RIM101—Regulator of IME2	In vitro	Biofilm formation	[23,24]
ROB1—Recombinant Beta-lactamase 1	In vitro/in vivo	Biofilm formation	[23]
TEC1—Thermoelectric cooler	In vitro/in vivo	Biofilm integrity	[25]
GCN4—General control transcription factor 4	In vitro	Biofilm metabolism	[26]
GAL4—Galactose-responsive transcription factor 4; RFX2—Regulatory factor X2	In vitro/in vivo	Negative regulator of biofilm formation	[24]
BCR1—Biofilm and Cell Wall Regulator 1	In vitro/in vivo	Regulator of three key families of *C. albicans*	[27,28]
ACE2—Angiotensin-converting enzyme 2; ADA2—Adenosine deaminase 2; ARG81—Arginine metabolism-regulation protein II; CAS5—CRISPR-associated protein; CRZ2—Coastal regulation zone 2; CZF1—Zinc cluster transcription factor 1; DAL81—DNA-binding transcription factor; FCR3—Fluconazole resistance protein 3; FGR27—Filamentous growth regulator 27; LEU3—Regulator of leucine biosynthesis; MET4—Microbial ecosystem therapeutic 4; NOT3—Negative regulator of transcription subunit; SNF5—Sucrose non-fermenting gene number 5; SUC1—Sucrose-proton symporter; TAF14—TBP-associated factor 14; TRY(2–6)—Transcriptional regulator of yeast form adherence; UGA33—Transcription factor for C6 zinc finger class; WAR1—Yeast zinc-finger transcription factor; ZCF (28,31,34,39,8)—Zn cluster family; ZFU2—Zn(II)2Cys6 transcription factor; ZNC1—Zinc cluster transcription factor; AHR1—Aryl hydrocarbon receptor	In vitro	Adhesion	[29,30]
ACE2—Angiotensin-converting enzyme 2	In vivo	Adhesion and hyphae formation	[31]
EFG1—Enhanced filamentous growth protein 1, SNF5	In vitro/in vivo	Adhesion and hyphae formation	[23,32]
MSS11—Multicopy suppressor of STA genes	In vitro	Hyphae formation	[33]
NDT80—Non-DiTyrosine 80	In vitro/in vivo	Hyphae formation and positive regulator	[34]
NRG1—Neuregulin 1; TYE7—Transcriptional activator of glycolytic gene expression	In vitro	Negative regulator of hyphae formation	[22,35]
STP2—Single touch payroll 2	In vitro	Adhesion, hyphae formation, and metabolism	[36]
RLM1—MADS-box transcription factor	In vitro/in vivo	Extracellular matrix organization	[37]
ZAP1—Zona pellucida gene-activating protein-1	In vitro/in vivo	Negative regulator of hyphae formation and extracellular matrix formation	[38]
UME6—Unscheduled meiotic gene expression	In vitro	Hyphae regulation and dispersion	[35]

**Table 2 microorganisms-12-02138-t002:** Extracellular signals responsible for *C. albicans*—bacteria biofilm formation.

Molecules	Oral Bacteria	Impact on *C. albicans*	References
Trans-2-decenoic (SDSF)	*S.mutans*, mitis group streptococcus	Inhibit candidal hyphae formation	[58]
Competence-stimulating peptide (CSP)	*S.mutans*	Inhibit candidal germ tube and hyphae formation	[59]
Membrane vesicles	*S.mutans*	Affect sucrose metabolism	[60]
Autoinducer-2	*S. gordonii*	Reduce farnesol effects on candidal hyphae formation	[54]
Hydrogen peroxide	*S. gordonii*	Promote candidal filamentous growth, and oxidative stress	
Farnesol	*C. albicans*	Low level: Increase bacterial growth High level: Inhibit *S. mutans* growth	[61]
3-oxo-C12-homoserine lactose (C12AHL)	*P. aeruginosa*	Kill *C. albicans* hyphae	[20]

**Table 3 microorganisms-12-02138-t003:** The virulence potential of *C. albicans* in the saliva of patients with different oral diseases.

Oral Disease	Total No. of Sample	Sample Source	Detection Method	*C. albicans* Prevalence	Inference	References
Dental caries	1393	Saliva	Sabourads Dextrose Agar	24%	Salivary *S. mutans*, *lactobacilli*, and *C. albicans* are present in caries	[67]
Dental caries	108	Saliva	CHROMagar Candida, Sabouraud agar	55.4%	Ecologic niche for *C. albicans* in carious teeth	[68]
Early childhood caries	14	Saliva	CHROMagar Candida	43%	Microbial diversity in early childhood caries	[69]
Dental caries		Saliva	Sabouraud agar with tetracycline, CHROMagar Candida	63%	*C. albicans* is related to the severity of caries	[70]
Dental caries	35	Saliva	CHROMagar Candida	77%	High abundance of *C. albicans* in caries and most strains are genetically related	[71]
Early childhood caries	40	Saliva	CHROMagar Candida	100%	Highly increased *C. albicans* abundance in severe early childhood caries	[72]
Dental caries	160	Oral rinse	Sabouraud agar culture medium	30.6%	Supports the etiological role of Candida in dental cariogenesis	[73]
Chronic and acute periodontitis	74	Saliva	Biochemical and CHROMagar Candida	87.5%	*C. albicans* is capable of colonizing in the periodontal pockets of chronic periodontitis patients	[74]
Periodontitis with diabetes	30	Saliva	CHROMagar Candida	53%	Increases the propensity of *C. albicans* in patients with periodontitis and uncontrolled diabetes	[75]
Chronic periodontitis with diabetes	30	Whole saliva	Biochemical tests	60%	High occurrence of Candida species in the saliva of diabetes patients with chronic periodontitis	[76]
Periodontitis with HIV	73	Saliva	CHROMagar Candida	79%	Candida spp carriage was increased in periodontally affected HIV-infected patients	[77]
Chronic periodontitis with diabetes	100	Whole saliva	Sabouraud agar	51.9%	A positive association between poor glycemic control and the prevalence of high candidal carriage	[78]
Chronic periodontitis	155	Oral rinse	CHROMagar Candida	42%	Higher rate of Candida colonization in periodontitis	[79]
Oral precancer and oral cancer	200/97	Saliva	Sabouraud’s dextrose agar	58% 72.2%	Oral precancer and cancer patients showed a higher number of *C. albicans* colonies	[80]
Oral cancer	52	Oral rinse	CHROMagar Candida	52%	Association between oral cancer occurrence and Candida oral colonization; *C. albicans* has a role in oral carcinogenesis	[81]
Oral cancer	100	Saliva	CHROMagar Candida	56.6%	Higher proteolytic activity affects the virulence of *C. albicans* in oral cancer patients	[82]

## Data Availability

The data and articles presented in this study are mentioned in the References.

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
