# Peer review of "Candida albicans Induces Oral Microbial Dysbiosis and Promotes Oral Diseases"

_microorganisms, 2024, doi:10.3390/microorganisms12112138_

Round 1
Reviewer 1 Report
Comments and Suggestions for Authors
The review article provides a comprehensive overview of the pathogenic mechanisms of Candida albicans in the oral cavity. While the scope of the review is commendable, there are several aspects that require significant improvement to enhance clarity, focus, and depth. My detailed comments are as follows:
1. The current structure of the review tends to list the conclusions of various studies without a clear focus or logical progression. Each section appears somewhat disjointed, lacking coherence and prioritization of key findings. The authors should provide a more in-depth analysis of their findings. They should highlight the most interesting results reported in certain papers in comparison to other results in other reports to give the reader a better understanding of the actual activity.
2. The discussion remains quite superficial. Instead of merely summarizing the findings of previous studies, the authors should delve deeper into the areas of consensus and controversy within the field. Particularly, the authors should offer their own insights or interpretations. This would enrich the discussion and demonstrate a more critical understanding of the topic.
3. Sections 4 and 5 have substantial overlap in their content. I suggest that the authors merge these two sections, carefully screen the studies covered and removing redundant information. This would streamline the review and avoid unnecessary repetition.
4. The language used in the manuscript is overly complex in some areas. Simplifying the text and using clear, concise expressions would help emphasize the key messages.
5. The references are somewhat outdated, with only 37 out of 131 citations from the last five years. Considering the rapid advancements in this field, I recommend incorporating more recent studies to ensure the review reflects the latest research developments.
6. Page 8, line 206, “S. mutants” was wrongly spelled.
7. C. albicans in some places were not in italics.
In conclusion, while the review provides a broad coverage of the topic, substantial revisions are needed to enhance its clarity, focus, and depth. Addressing these issues will significantly improve the manuscript’s impact and readability.
Comments on the Quality of English Language
The quality of English language is acceptable.
Author Response
Attached below

Reviewer 2 Report
Comments and Suggestions for Authors
In the review titled “Candida albicans induces oral microbial dysbiosis and promotes oral 2 diseases”, the authors aim at s the mutualistic interaction of C. albicans in oral biofilm formation 20 and polymicrobial interactions in oral diseases. The article is well written and captures major highlights of oral interaction of Candida with other microbial species and their health consequences on the host. There are few changes that the authors need to make for the acceptance of this well-organized review article.
1. Sentence 60: The authors mention that “This narrative review focuses on C. albicans changes in the oral cavity and its mutualistic interactions with oral commensal bacteria”. However, it is not clear what these changes refer to and are these in response to a set of criteria that the authors forgot to mention (e.g age, gender, life style etc). The focus of the review needs to be stated clearly and elaborated properly.
2. There are few occasions where C.albicans needs to be italicized.
3. It will be very helpful for the readers if the authors could make a model figure of various affects of candida on the host (e.g inflammatory damage, pre-cancer etc) and indicating key players of such effects (e.g the cytokines involved).
4. The authors did include various transcription factors that regulate the biofilm organization. However, later in the review and under discussion, they did not highlight/discuss the effect of conditions like pre-cancer, peri-dontal disease, dental caries) on the expression/regulation of such transcription factors
Author Response
Attached below.

Round 2
Reviewer 1 Report
Comments and Suggestions for Authors
The revised manuscript shows some improvements in language and logic; however, the authors should carefully verify the references. I have noticed that Reference 90 is cited in the manuscript (on page 9, lines 228-230 and 232-236) to describe changes in Candida albicans carbohydrate metabolism when co-cultured with Streptococcus mutans. However, Reference 90 (“Costa Oliveira, B. E.; Ricomini Filho, A. P.; Burne, R. A.; Zeng, L. The Route of Sucrose Utilization by Streptococcus mutans Affects Intracellular Polysaccharide Metabolism. Frontiers in Microbiology, 2021, 12, 636”) does not mention anything about Candida albicans. Please ensure that all references are checked thoroughly to avoid similar discrepancies.
After the revision, I believe the manuscript is ready to be published.
Author Response
Attached below
